# Growth and Safety Assessment of Feed Streams for Black Soldier Fly Larvae: A Case Study with Aquaculture Sludge

**DOI:** 10.3390/ani9040189

**Published:** 2019-04-23

**Authors:** Eric Schmitt, Ikram Belghit, Johan Johansen, Raymond Leushuis, Erik-Jan Lock, Diede Melsen, Ram Kathirampatti Ramasamy Shanmugam, Joop Van Loon, Aman Paul

**Affiliations:** 1Protix B.V. Industriestraat 3, 5107 Dongen, The Netherlands; Raymond.Leushuis@protix.eu (R.L.); diedemelsen@hotmail.com (D.M.); Aman.Paul@protix.eu (A.P.); 2Institute of Marine Research, P.O. Box 1870 Nordnes, 5817 Bergen, Norway; Ikram.Belghit@hi.no (I.B.); Erik-Jan.Lock@hi.no (E.-J.L.); 3Salten Havbrukspark, 8120 Nygårdsjøen, Norway; johan@havbruksparken.no; 4Laboratory of Entomology, Wageningen University, P.O. Box 16, 6700 AA Wageningen, The Netherlands; krsram_91@yahoo.co.in (R.K.R.S.); joop.vanloon@wur.nl (J.V.L.)

**Keywords:** black soldier fly larvae, sludge, mineral composition, safety risks, aquaculture

## Abstract

**Simple Summary:**

The production of food is an intensive source of environmental impact. In aquaculture, one source of impact is solid waste, which contains high concentrations of minerals, other nutrients, and metals. The larvae of *Hermetia illucens* are capable of consuming this material, but applying technology that is based on these larvae for managing waste streams, like those from aquaculture, requires careful examination of safety risks. A study is performed examining the growth performance of larvae that were fed on solid aquaculture waste. Then, a thorough analysis of safety risks from inorganics is performed to serve as a guideline for how to assess the safety of waste streams, such as these. The practitioner can use this as a template for the safety assessment for other high risk organic streams as feed for larvae.

**Abstract:**

The production of food is an intensive source of environmental impact. In aquaculture, one source of impact is solid waste, which contains high concentrations of minerals, other nutrients, and metals. The larvae of *Hermetia illucens* are capable of consuming this material, but applying technology that is based on these larvae for managing waste streams, like those from aquaculture, requires careful examination of safety risks. A study is performed examining the growth performance of larvae that were fed on solid aquaculture waste. Subsequently, a thorough analysis of safety risks from inorganics, with detailed the results on microelements that have previously received little attention in the literature, is performed to serve as a guideline for how to assess the safety of waste streams such as these. Findings confirm existing results in the literature that Cd is bioaccumulative, but also that other elements, including Hg, Mn, and especially K, are bioaccumulative. To the authors’ knowledge, this is the first research where the accumulation of Ag is also tested. The results of these tests are explained within the context of regulations in various countries where *Hermetia illucens* is cultivated, serving as a reference for practitioners to rigorously screen out high risk feed streams that they may consider using as feed sources. It is intended that these references and the demonstrated accumulation of a range of elements motivate comprehensive industry safety practices when evaluating new feed sources.

## 1. Introduction

Aquaculture is a rapidly growing link in the food chain [1,2], and waste output from this industry is also rising. Between 1979 and 2013, the value of the trade in salmonoids has increased at an average rate of 10% and, in 2018, salmonoid species made up a large share of worldwide production in quantity (7.4%) and value (18.1%) [2]. Solid waste is particularly problematic during the production of Atlantic salmon (*Salmon salar* L.). Fish only absorb and digest 10 and 50% of the phosphorus and nitrogen available in the feeds, respectively. Aquaculture wastewater contains organic matter; uneaten fish feeds and fish faeces; and, dissolved inorganic nutrients, like excreted ammonia, nitrite, nitrate, and phosphorus compounds. In open sea farming systems, such as those that are used in most salmon production units, these nutrients circulate in the aquaculture system, and they can accumulate to toxic levels that are harmful to fish and other marine organisms and result in eutrophication [3]. According to the report that was published by Alanära et al. (2013), for every tonne of wet fish or approximately 300kg of dry fish produced, approximately 150 kg solid waste is obtained [4].

A number of methods exist to remediate aquaculture waste. Recirculating aquaculture systems (RASs) collect and remove waste products from the pens and reuse the water by pushing it back into the pens [3]. A RAS can be designed to manage nitrogenous wastes and gaseous exchange, nutrients, reduce toxicity, among other things. However, RAS are not well suited in managing fine solid wastes [5]. Furthermore, they are suitable for closed aquaculture systems, but are less appropriate for open-sea farming; there, some eutrophication can occur, but nutrient loss, especially P, can be a major issue. Another emerging technique that reduces the presence of nutrients in water is aquaponics, which is a combination of aquaculture and hydroponics. Nutrient-rich water from the aquaculture pens is used to grow plants in the hydroponic system. However, this method also focuses on using the nutrients from the water and not on the nutrients that are present in the solid wastes produced in the aquaculture pens [3]. New methods to treat solid aquaculture wastes (SAWs) require further development.

Black soldier fly (*Hermetia illucens* L.) larvae are capable of eating a wide range of organic materials, which has attracted interest in them as a means of reducing the impact of waste organics and recapturing the nutrients in them by extracting high quality proteins and lipids from the harvested larvae [6,7]. Research on the application of black soldier fly larvae (BSFL) began with a focus on remediation of waste products, such as manure [8,9,10,11,12,13,14]. In recent years, many new companies that produce food and feed ingredients from BSFL have been initiated [5]. Many trials have been conducted to valorise BSFL lipids and proteins in animal feed, including fish species that are grown in systems that produce SAWs, which indicate that these nutrients are high quality and can replace other sources, such as soy and fishmeal [15,16,17,18,19]. Researchers now focus on a wide range of potential feed sources for BSFL in order to determine the best diets for the insects and to uncover the full waste remediation potential of the technologies that developed around them [20,21]. However, the possibilities of utilizing SAW to feed BSFL and the safety of the outputs remain unstudied.

The European Food Safety Authority suggested that BSFL are at a risk of carrying biological and/or chemical food safety hazard as a function of feeding substrate [22]. Further, they indicated that targeted thermal processing could eliminate biological hazards, such as pathogenic biota (macro and micro). However, chemical hazards, like heavy metals, toxins, etc., are not always possible to eliminate in this fashion [22]. Therefore, companies that are involved in the business of mass-rearing these insects should make a careful choice of feed substrate. Campo et al. (2010) indicated that SAW could carry a wide range of chemical hazards, especially heavy metals, which could have harmful effects on biological life [23]. Another topic that remains to be investigated is the accumulation of these hazardous chemicals in BSFL. With the aim to valorise SAW as potential feedstock for BSFL, this research was conducted with the following objectives:Investigate the nutritive potential of SAW-based diets for BSFL.Investigate the accumulation of hazardous chemicals (heavy metals) in BSFL fed with SAW based diets.

## 2. Materials and Methods

### 2.1. Experimental Set-Up

The test diets were mixtures of SAWs and water. The SAW used in the study was obtained from Helgeland Smolt’s young salmon molting facility based in Reppen, Norway. It consisted of the reject water from the drum filter of the water recirculation system of aquaculture tanks. This SAW was processed at Gildeskal Research Station AS (GIFAS,) according to details that are mentioned in Figure 1. Dewatered sludge was dried and sanitised in an oven using superheated steam at 250–300 °C, which increased the temperature of the SAW to 100 °C. At the end of each drying cycle, no more dewatered sludge was added for approximately 15 minutes, resulting in a steady reduction in relative humidity (RH) in the oven, while the SAW temperature increased to 115 °C. For SAW95 and SAW90, we stopped at 45% and 60% RH, respectively, to evaluate the impact on protein degradation and subsequent digestibility. Figure 1 mentions the crude protein, lipid, and ash content of the two SAWs used for feeding studies. Figure 2 mentions the mineral composition of the two SAWs. Table 1 shows the proximate composition of the SAWs.

Upon reception of SAW 90 and SAW 95 at Protix (Dongen, NL), they were examined for physical characteristics, such as texture, colour, and smell. Both of the ingredients were then standardized to 30% dry matter by the addition of appropriate amounts of lukewarm tap water (40 ˚C) before feeding.

Eight replicates per recipe were tested during the study. The test crates were placed in trolleys and arranged according to the experimental design in Appendix A to address the possible positional effects on performance. The test was only conducted once based on the outcomes of preliminary testing. The outcomes were deemed to be consistent with the preliminary findings and in line with literature on similar materials, which showed low growth on the feed streams derived from manure streams [8,9,10,11,12].

### 2.2. Rearing of BSF Larvae

Larvae used in the production process were sourced from Protix. These larvae were approximately eight days old (post hatching) and had an average fresh body weight of 14 mg. Larvae with these characteristics were used on the basis of preliminary investigations. Prior to the start of the tests, the larvae were fed a diet composed of from plant derived ingredients. The experiments were conducted in containers with dimensions of 40 cm x 30 cm x 11 cm, where each container accommodated approximately 1.25 × 10^4^ larvae.

Only sanitized sludges (SAW 90 and SAW 95) and water were used to develop the feed mix. The feeding recipes and regimes that were used in the study were developed using preliminary testing referred to above and they are mentioned in Appendix A. 

The subsamples of larvae from each container were sampled several times during testing for the estimation of average larval weight and number of larvae in containers to monitor survival. The prior measures growth rate of larval body mass, while the latter quantifies dosing accuracy and mortality. Sampling and measurement (S) was performed, as follows:S1. Subtract the plastic weight of the container from the total weight of the container including plastic and biomass to obtain biomass in each container (g).S2. Thoroughly homogenize the content of container.S3. During stirring, take a sample of substrate and at least 50 larvae.S4. Weigh this sample to get the sample weight (g).S5. Count the number of larvae in sample.S6. Weigh the larvae.S7. Calculate the average larval weight is calculated by dividing outcomes of S4 by outcomes of S5. In other words S4/S5S8: The number of larvae in the container is estimated as dividing outcomes of S5 by outcomes of S2 and multiplying this number with outcomes of S1. In other words: (S5/S2)∙S1

At the end of feeding, the larvae were sieved out from the mix of frass and feed residue to obtain total larval biomass in kilograms by weighing the total cleaned larvae from a growing container. This final fresh larval biomass is referred to as the yield of the container. Larvae that were fed on a non-SAW diet were not grown in this research project, but a control larvae group for comparison purposes from an earlier research project with Protix and NIFES will be used as reference [24]. The control feed consisted of a plant-based material that consisted of processed wheat.

### 2.3. Compositional Analyses

All of the compositional analyses were realized on SAW 90 and SAW 95, harvested larvae and frass and feed residue samples. The dry matter of samples was analysed by first freeze drying the materials at −20 °C for 72 h and then allowing them to reach 25 °C in vacuum (until constant weight was achieved).

Total nitrogen estimation was undertaken on freeze dried, ground samples using a CHNS elemental analyser (Vario Macro Cube, Elementar Analysensysteme GmbH, Germany) and quantified using the N-to protein factor of 6.25 [25]. The instrument was calibrated with ethylene diamine tetra acetic acid (Leco Corporation, USA). Sulfanilamide (Alfa Aesar GmbH & Co., Germany) and a standard meat reference material (SMRD 2000, LGC Standards, UK) were used as the control samples.

Total lipids (ethyl acetate extraction) were gravimetrically estimated in wet ground samples. The ash content of wet ground samples was investigated by heating the samples at 550 °C overnight in a muffle furnace (Thermolyne F 30,430 CM, USA). Carbohydrate content inclusive of fibre content was calculated by mass difference according to the formula that was described by [26]: % Carbohydrates = 100 − (% Crude Protein + % Fat + % Water + % Ash). The mineral composition of freeze-dried ground samples was analysed by inductive coupled plasma spectrometry (ICP-MS) after wet digestion in a microwave oven, as described by [27], with some modifications. In brief, the samples were digested using 69 % nitric acid (2 mL) and 30 % hydrogen peroxide (0.5 mL) using a microwave digester (UltraWave, Milestone, Italy). The resultant solutions were diluted to 25 mL with deionized water (MilliQ, Merck Millipore, USA). Mineral concentrations in the samples were quantified by ICP-MS (iCapQ ICPMS, ThermoFisher, USA) that was equipped with an autosampler (FAST SC-4Q DX, Elemental Scientific, USA). Data were collected and processed using the Qtegra ICPMS software (ThermoFisher Scientific, USA). 

### 2.4. Bioaccumulation

The accumulation of elements in the larvae is a major concern when studying feed conversion. Accumulation of desirable nutrients is potentially beneficial in applications of the larvae, while the accumulation of others, such as heavy metals, is undesirable. Following [28], the bioaccumulation factor (BAF) was calculated on a dry matter (DM) basis, as BAF = concentration in the organism (DM-basis)/concentration in the feed provided (DM-basis). A BAF that is larger than 1 implies the bioaccumulation of the element from the feed into the insect [28].

BSFL can accumulate high levels of calcium and manganese, and they contain a relatively high amount of calcium [6]. Liland et al. (2017) demonstrated the linkage between dietary calcium and accumulation [24]. They found that an increase in dietary calcium was directly linked to increased calcium accumulation in BSFL due to its future role in pupation [29]. Manganese has several important functions in insect physiology, including a role as metal co-factor for many cellular functions [30].

### 2.5. Statistical analyses

Statistical analysis was performed to compare the growth performance of the two feeds, and to determine whether the accumulation of contaminants exceeds the regulatory limits. In both cases, versions of linear regression are used to obtain t-tests.

To compare the performance of the larvae on the two SAW materials, simple linear regressions were performed, where the dependent variable is the variable of interest, size, for example, and the independent variable is an indicator variable for the material on which the larvae fed.

It is known that the larvae of BSF accumulate certain heavy metals [23]. The feed that was used in these trials contains a number of them in non-negligible quantities. It is essential when exploring feed streams with this risk profile to confirm whether the larvae grown on them accumulate heavy metals beyond regulatory limits for feed, which are foremost based on safety concerns. In addition to heavy metals, regulatory limits also exist for micronutrients. The question to be answered is whether quantity of heavy metals and micronutrients per gram dry matter exceeds regulatory limits. A regression approach is taken to answer this question. This is explained now in some detail, so that other practitioners can perform this type of analysis during their own feed tests. 

First, given a dataset with *N* rows indexed by *i,* we define the variables Concentration (mg/g DM-regulatory limit), Feed (type of feed material), Experimental unit (the unique growing container), and EL (type of element (heavy metal or micronutrient) for *k* types of elements). Note that EL is a dummy variable taking values 0 or 1, depending on whether the row *i* corresponds to a specific metal. Similarly, Feed is a dummy variable for whether the feed is SAW90 or SAW95. These are input for the following mixed-effects regression model: Concentration = Σ_j_^k^ β_j_ EL_ki_ + β_k+1_Feed_i_ + π_i_(Experimental Unit) + ε_i_(1)

There are a few differences emerging from a typical linear regression. The first is that there is no intercept. This is because we desire the expected values for the effects of each of the heavy metals and micronutrients measured. Since the concentration response is the measured value minus the limit, the *p*-value for these effects are for the null hypothesis that there is no difference between the measured value and the limit. Setting up the model this way lets us test whether the larval contents are statistically over and/or under the limits using one-sided or two-sided t-tests. Another difference from a classic regression model is the random effect estimated with the π values, which are random intercepts. These are used to adjust the model for metal and micronutrient concentration variability between the crates that could be driven by between-crate variability, such as different growth conditions or other factors rather than the metal accumulation process. The mixed-effects models that are estimated in this paper were computed using the *lmerTest* package and dependencies in the R statistical software version 3.4.4 [31,32].

## 3. Results and Discussion

### 3.1. Growth Performance

The feeding trials were conducted over a period of ten days. Larvae were sampled and weighed at the start of the test (Day 0) and on Days 2, 3, 4, 5, 6, 7, 8, 9, and 10, respectively. On Day 10, the larval and residue yield were also calculated. Figure 3 shows the growth curves for 16 crates studied in the present work. Growth of larvae fed with two SAW mixes was more or less the same before day 7. On Days 7, 8, and 9, the biomass of larvae fed on SAW 90 mix reached higher values than that of the larvae fed SAW 95 mix. From Figure 3, it is also visible that the average larval weights fed with two recipes diverge the most on Day 9. The difference in average weight on Day 9 was estimated using linear regression, showing that SAW 90 mix fed larvae were heavier than SAW 95 mix fed larvae by approximately 0.012 g (*p*-value < 0.001). On Day 8, the difference was also significant (*p*-value < 0.001), but smaller at 0.009 g. These results might be due to the crude protein content or if the protein conversion factor is inaccurate then the nitrogen present in the feeding media, which was higher in SAW 90 (21.4% of DM) than SAW 95 (12.9% of DM). Between Day 9 and 10, the larvae are not fed, which is responsible for general weight loss. This phenomenon also results in the re-convergence of the weights of larvae from two SAW mixes. This weight loss is due to dehydration and starvation. Pupation had begun, at which point the animals stop eating. 

Number of larvae in each container was also estimated on multiple days. Figure 4 shows the estimation of these larvae numbers, showing their number per container. The black horizontal line shows the targeted population size of each container. At the first in-test measurements on Day 2, 3, and 4, we see a slight increase in population from under target line to above target line. Obviously, the population could not have increased; this increase is observed due to the increase in weight of larvae, which improves the ease of measurement. Approximately 10% and 20% decreases in population were observed on Day 8 and 9, respectively. The reason behind this is not known. On Day 10, the average mortality rate that was observed for SAW90 was 9.1% (SD 10.79%), while the mortality rate for SAW 95 was 15.4% (SD 16.28%). According to the Wilcoxon rank-sum test, we do not reject the null hypothesis that these mortality rates differ (*p*-value = 0.6454).

The crates that were fed SAW95 had an average larval yield of 0.57kg (SD 0.13kg) and the crates fed SAW90 had an average larval yield of 0.62kg (SD 0.1kg). Dry matter larval yields were measured at the end of the test. Consistent with the average larval weight measurement, the containers in which larvae were fed with SAW 90 mix achieved higher yields. Dry matter conversion has been calculated as the difference between the end and start weight (kg) of the larvae (fresh larval biomass at the start of rearing), divided by the kilograms of dry matter fed throughout the experiment. The average dry matter conversion rate is 11.9 (SD = 3.1) and 14.8 (SD = 5.7) for the SAW 90 and SAW 95 fed containers, respectively. The crates fed SAW95 had an average residue yield of 2.36kg (SD 0.23kg) and the crates fed SAW90 had an average residue yield of 2.15kg (SD 0.07kg). Thus, in both cases residue yield was approximately 3.7 times more than the wet larval yield.

### 3.2. Composition Analyses

Table 2 shows the dry matter, crude protein, lipid, ash, and carbohydrate contents of the larvae. The statistical difference between the composition of larvae obtained using two feed mixes was analysed using the Wilcoxon rank-sum test, with only the dry matter showing a statistically significant difference (*p*-value = 0.003). In the current study, the crude protein content of the larvae fed with two SAW mixes (~ 40% of DM) were similar to previously published data of BSFL [6], but lipid content was significantly lower (~ 7% of DM) as compared to the commonly reported lipid composition of BSFL (~ 25–30% of DM) [6]. Lipid accumulation in insects varies with exact life stage, diet, and environment, and these larvae may have lower lipid amounts due to the lack of available lipids and carbohydrates in the diet, which also likely remains a reason for their low growth [29]. It has been already indicated in Figure 2 that both SAW ingredients have substantial amounts of cadmium, copper, and zinc. Lindqvist et al. (2001) and Ortel et al. (1995) describe how high levels of dietary metals, including cadmium, copper, and zinc are linked to reduced fat accumulation in insects [30,33]. It could be that the low availability of nutrients, such as lipids and carbohydrates, increases the accumulation of heavy metals and other elements, but additional research is needed to confirm this hypothesis.

The mineral composition of the two SAW mixes and larvae that were fed with two SAW mixes as well as a control feed are mentioned in Table 3 and Table 4, respectively. Trace metals, such as vanadium, chromium, nickel, arsenic, and mercury have previously been analyzed in BSFL [34,35,36,37]. However, to the best of our knowledge, this is the first report indicating the presence (or accumulation) of silver in BSFL, though silver was present at concentrations of just a few micrograms per kilogram. Calcium was the macro-mineral present in highest concentrations in larvae fed with both SAW mixes, while manganese was the micro-mineral that was present in a highest concentration. Both calcium and manganese were also the major macro-mineral and trace mineral present in the two SAW materials, respectively (see Figure 2). 

When evaluating mineral accumulation with regards to regulations, three types of limits (or levels) have been taken into account: (1) Rejection limits; (2) Action limits; and, (3) Maximum tolerance levels. The rejection limits or maximum content could be defined as ‘maximum levels in mg/kg of the feed material or compound feeds, based on a moisture content of 12%, unless mentioned differently. If this limit is exceeded, then the product is not suitable for use as feed material or animal feed’ [38,39,40]. Action limit is a term that is commonly used in GMP+ (most popular animal feed and pet food quality management systems) documents. This limit is defined in agreement with the sector, supplier, or customer. It could be defined as ‘maximum levels in mg/kg of feed materials or compound feeds, based on a moisture content of 12%, unless differently mentioned. If this limit is exceeded, then an investigation into the cause should be undertaken and corrective measures should be taken to remove or control the cause’ [40]. Maximum tolerance levels are more strict in comparison to the rejection limits or actions limits and could be defined as ‘dietary levels that, when fed for a defined period of time, will not impair animal health or performance’ [41].

Table 5 provides the maximum tolerance levels (based on chronic consumption), action limits, and rejection limits of minerals that were considered in this study. The maximum tolerance levels that were used in this study were according to the values estimated for poultry [38], because live larvae are already approved for poultry consumption in Europe [42]. When required, the data was converted into dry matter basis, so that all measurements are in the same units.

Table 6 indicates the comparison of rejection limits with the amounts detected in larvae. The comparison is most effectively made using one-sided t-tests, the outcomes of which are provided, to test whether the concentration of an element in the larvae exceeded this limit. The one-sided t-test is efficient in this case, because we are concerned with whether the concentrations are too high. The action limits were not compared with the obtained results, because it was clearly visible that larvae were below the action limits. Both larvae fed with SAW 90 and 95 mixes were detected to contain arsenic (As) in concentrations that were above rejection limits proposed by the European Commission [39]. The inorganic form of arsenic, particularly the trivalent arsenic (arsenite), is known to be extremely toxic during chronic exposure [43]. Chronic dietary exposure to arsenic may result in the development of peripheral neuropathy, modifications in normal heme metabolism, impairment of renal function, and other serious disorders in animals [44]. Larvae with such levels of arsenic cannot be used for direct feeding to poultry animals in Europe (Table 5). However, it may be possible to process them into insect protein meal, which could be included in compound animal feeds, such that the total amount of arsenic in compound feed is below the European Union (EU) rejection limits, since the insect meal might be a low enough portion of the total mix that it is not likely that an animal consuming it will intake a large dose.

Table 7 shows the comparison of maximum tolerance levels of minerals with the amounts detected in larvae. To our surprise, larvae were found to contain calcium, potassium, magnesium, and phosphorus in amounts exceeding the maximum tolerance levels. In normal concentrations, these minerals have an important role in animal nutrition [45]. However, repeated dietary exposure to these minerals in such concentrations could negatively affect animal health. In pets animals, like dogs, a high level of dietary calcium (hypercalcemia) is responsible for neoplasia, primary perparathyroidism, chronic kidney disease, and other serious health issues [46]. On the other hand, high levels of potassium, magnesium, and phosphorus in animals diets is associated with issues, such as chronic kidney disease, blood electrolyte abnormalities, and increased risk of cardiovascular diseases, respectively [41,42,44].

### 3.3. Bioaccumulation

Table 8 displays the bioaccumulation factors of the elements in the feed and their standard deviation. The bioaccumulation of the heavy metals As, Cd, and Pb was previously studied in [28]. The results that are reported here are not entirely consistent with those reported in [28]. Both the present work and [28] concur that Cd is highly accumulative. Divalent Cd cation has ionic radii that are similar to the divalent Ca cation, which makes the entry of cadmium in animal cells easy via calcium channels without any dependency on endocytosis or ATP [47].

As is shown to accumulate when larvae fed on SAW90 but not on SAW95 and the factors are higher than that reported previously for BSF (0.49–0.58 [28]). Studies on another dipteran larva (*Chironomus riparius*) have shown dose-dependent accumulation of As with feed. Moreover, it was suggested that *C. riparius* larvae could biotransform the arsenate form into the arsenite form as a part of detoxification metabolism. In the same study, it was also found that pupae contain lower amounts of As in comparison to larvae, indicating that larvae excrete substantial amounts of As before transforming into pupae [48]. Pb does not appear to accumulate, whereas bioaccumulation factors that are higher than 1 have been reported (1.1–1.8 [28]) for this metal.

Additionally, Hg is also found to be bioaccumulative in larvae that were fed on the SAW diets. Dipteran larvae are also known to bioaccumulate Hg. A study indicated that aquatic dipteran larvae can accumulate more Hg in comparison to their adults [49], suggesting that bioaccumulation of Hg is linked to the life cycle of these insects. The results of the present study show how the bioaccumulation of these three heavy metals might manifest in a heavy metal rich, real-world feed. In addition to heavy metals, Table 8 reports the bioaccumulation of other elements. The accumulation of many of these elements have been studied in Liland et al. (2017), where larvae were also fed on a sea-sourced diet anda number of these are reported to be bioaccumulative [24]. This is also the case in the present study, in which K shows particularly high accumulation.

This paper studies the performance and safety of SAW materials. However, it is crucial to expand studies in order to provide general guidelines for assessing the safety of different feeds and BSFL produced on them also for other feeds. In this research, we show that, in addition to well-known contaminants of BSFL, such as heavy metals, micronutrients can also reach levels that exceed regulatory norms. It may not be obvious to companies and practitioners which metals and micronutrients are likely to pose problems, so it is recommended that a broad screening of metals and micronutrients is performed on feeds for the BSFL that do not have known characteristics, especially if the feed material is likely to contain concentrated micronutrients. The paper provides a list of regulatory limits that can be used to evaluate the concentration of metals and micronutrients in the larvae. However, this is potentially incomplete and it will become out of date. Readers are encouraged to look to the most recent regulatory updates for current best practices.

## 4. Conclusions

This case study evaluates the growth potential of BSF larvae on SAW materials and it provides guidance to practitioners on how to comprehensively test for microelement contamination in the larvae. The results of this study show that SAW materials can be used to grow BSFL, but that these larvae will contain levels of heavy metals and micronutrients in excess of regulatory limits if larvae are only fed on SAW. Growth rates are lower than for other feed materials tested in the literature. In order to increase the output of a diet, including SAWs, other ingredients are necessary, and further research is needed to optimize such blends.

While the growth results are important, the main focus of this work is on safety. BSFL fed on a high SAW diet accumulate metals and micronutrients at levels that exceed regulatory limits. A general finding of this study is that many elements are capable of accumulating in the larvae. Existing literature, such as [28], have demonstrated that heavy metals, especially cadmium, can accumulate in the larvae of BSF. However, this study shows that many other elements not widely covered in the literature and are also possible sources of risk. It is likely that other elements or materials are capable of accumulation, so the authors recommend that practitioners considering using a new feed that is highly concentrated or likely to contain high concentrations of chemicals or microelements carefully scan for those risks in the feed and larvae that could exceed limits to ensure that those feed streams are being used safely. Once the level of accumulation of elements and chemicals in the larvae is known, further steps may still be possible to mitigate this effect, such as mixing the meal that is made from larvae in a larger blend that achieves concentrations of the elements and chemicals below regulation. However, this step must be performed with thorough attention to the identified risks.

## Figures and Tables

**Figure 1 animals-09-00189-f001:**
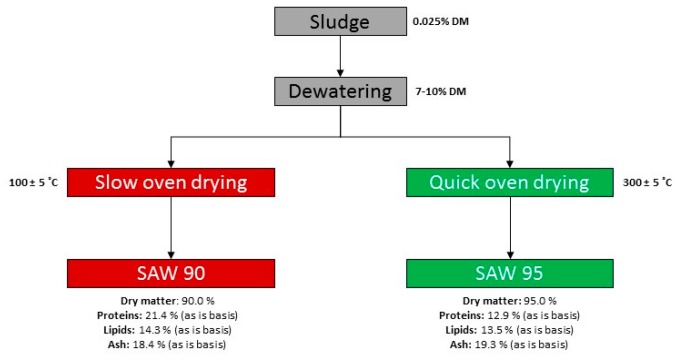
Method of producing sanitized solid aquaculture wastes (SAW) used for feeding insects.

**Figure 2 animals-09-00189-f002:**
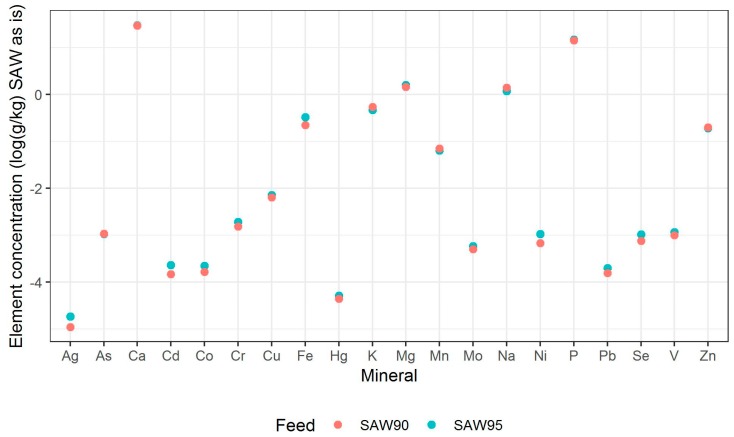
Mineral concentrations in log_10_(g/kg) of feed on an ‘as is’ basis.

**Figure 3 animals-09-00189-f003:**
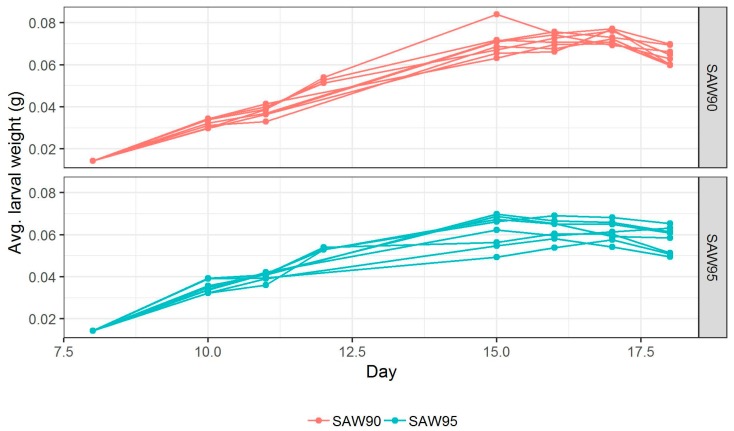
Growth performance of larvae fed with two SAW based diets.

**Figure 4 animals-09-00189-f004:**
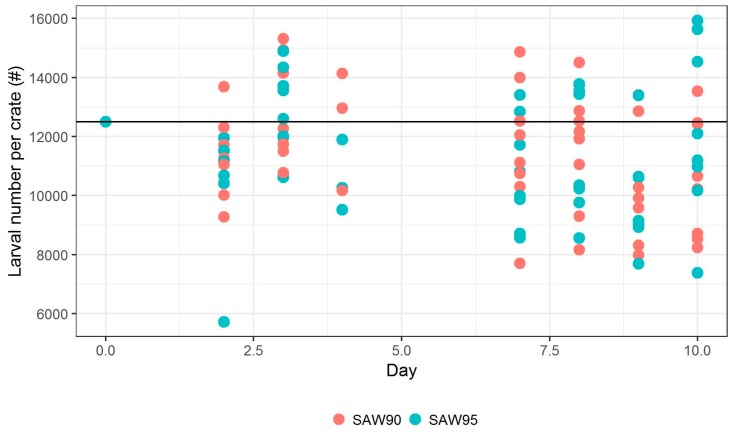
Number of larvae in test containers. The count on Day 0 is based on dosing estimation.

**Table 1 animals-09-00189-t001:** Proximate composition of two SAW-derived feed mixes and a control feed.

Component	Control [24]	SAW 90	SAW 95
**Dry matter**	31.1	42.7	43.8
**Crude proteins**	10.8	14.1	15.9
**Crude lipids**	4.8	6.6	6.7
**Ash**		4.8	4.9
**Carbohydrate**		17.3	16.3

**Table 2 animals-09-00189-t002:** Proximate composition of black soldier fly larvae grown on two SAW-derived feed mixes and a control feed.

Component	Control [24]	SAW 90	SAW 95
**Dry matter**	36.6 ± 0.50	36.7 ± 1.82	33.7 ± 1.35
**Crude proteins**	40.0 ± 0.90	41.8 ± 1.15	42.5 ± 1.46
**Crude lipids**	33.8 ± 1.60	7.2 ± 2.07	7.6 ± 2.14
**Ash**	5.1 ± 0.4	11.9 ± 2.83	10.1 ± 0.44
**Carbohydrate**	36.6 ± 0.50	39.1 ± 4.24	39.8 ± 2.94

SAW: Solid aquaculture waste; DM: dry matter, expressed as percentage of wet weight. Values for crude proteins, crude lipids, ash, and carbohydrates are expressed as percentages on dry matter basis. Values are mean ± standard deviation. Except for the dry matter, none of the of the components differ significantly between SAW90 and SAW95, *p* < 0.05; *n* = 8.

**Table 3 animals-09-00189-t003:** Mineral concentrations of feeding media used for the black soldier fly larvae growth trial.

	Control Feed [24]	SAW 90 Feed	SAW 95 Feed
**Ca ^1^**	2.4	82.9 ± 23.61	68.0 ± 3.10
**K ^1^**	10.0	9.9 ± 2.98	9.8 ± 1.20
**Mg ^1^**	1.8	4.1 ± 1.03	3.9 ± 0.18
**P ^1^**	4.6	19.2 ± 4.16	16.7 ± 2.95
**Na ^1^**	4.8	2.2 ± 0.39	1.9 ± 0.26
**Fe ^1^**	0.23	0.5 ± 0.06	0.6 ± 0.16
**Zn ^1^**	0.042	0.3 ± 0.04	0.2 ± 0.05
**Mn ^1^**	0.048	0.2 ± 0.01	0.2 ± 0.01
**V ^2^**	-	0.9 ± 0.30	0.8 ± 0.36
**Cr ^2^**	-	1.6 ± 0.43	1.7 ± 0.56
**Co ^2^**	-	0.2 ± 0.03	0.2 ± 0.06
**Ni ^2^**	-	0.7 ± 0.18	0.8 ± 0.30
**Cu ^2^**	6.3	12.7 ± 1.28	11.6 ± 1.53
**As ^2^**	-	3.3 ± 0.22	2.1 ± 0.24
**Se ^2^**	0.07	1.01 ± 0.16	1.2 ± 0.26
**Mo ^2^**	-	1.0 ± 0.13	0.8 ± 0.18
**Ag ^3^**	-	5.1 ± 4.14	13.1 ± 10.70
**Cd ^2^**	-	1.0 ± 0.10	1.4 ± 0.10
**Hg ^2^**	-	0.2 ± 0.01	0.2 ± 0.01
**Pb ^2^**	-	0.3 ± 0.05	0.3 ± 0.06

SAW: Solid aquaculture waste. Values are mean ± standard deviation; *n* = 8. ^1^ Values reported as g kg^−1^ DM. ^2^ Values reported as mg kg^−1^ DM. ^3^ Values reported as µg kg^−1^ DM.

**Table 4 animals-09-00189-t004:** Mineral concentrations black soldier fly larvae fed on different media.

	Control Larvae [24]	SAW 90 Larvae	SAW 95 Larvae
**Ca ^2^**	8.4 ± 0.4	82.9 ± 23.61	68.0 ± 3.10
**K ^1^**	10.2 ± 0.2	9.9 ± 2.98	9.8 ± 1.20
**Mg ^1^**	2.1 ± 0.1	4.1 ± 1.03	3.9 ± 0.18
**P ^1^**	6.8 ± 0.1	19.2 ± 4.16	16.7 ± 2.95
**Na ^1^**	1.0 ± 0.1	2.2 ± 0.39	1.9 ± 0.26
**Fe ^1^**	0.21± 0.02	0.5 ± 0.06	0.6 ± 0.16
**Zn ^1^**	68 ± 1.6	0.3 ± 0.04	0.2 ± 0.05
**Mn ^1^**	0.19 ± 0.01	0.2 ± 0.01	0.2 ± 0.01
**V ^2^**	-	0.9 ± 0.30	0.8 ± 0.36
**Cr ^2^**	-	1.6 ± 0.43	1.7 ± 0.56
**Co ^2^**	-	0.2 ± 0.03	0.2 ± 0.06
**Ni ^2^**	-	0.7 ± 0.18	0.8 ± 0.30
**Cu ^2^**	7.9 ± 0.2	12.7 ± 1.28	11.6 ± 1.53
**As ^2^**	-	3.3 ± 0.22	2.1 ± 0.24
**Se ^2^**	0.1 ± 0	1.01 ± 0.16	1.2 ± 0.26
**Mo ^2^**	-	1.0 ± 0.13	0.8 ± 0.18
**Ag ^3^**	-	5.1 ± 4.14	13.1 ± 10.70
**Cd ^2^**	-	1.0 ± 0.10	1.4 ± 0.10
**Hg ^2^**	-	0.2 ± 0.01	0.2 ± 0.01
**Pb ^2^**	-	0.3 ± 0.05	0.3 ± 0.06

SAW: Solid aquaculture waste. Values are mean ± standard deviation; *n* = 8. ^1^ Values reported as g kg^−1^ DM. ^2^ Values reported as mg kg^−1^ DM. ^3^ Values reported as µg kg^−1^ DM.

**Table 5 animals-09-00189-t005:** Rejection limits (RL), Action limits (AL), and Maximum tolerance levels (MTL) of detected minerals.

	RL [38,39]	(AL) [40]	MTL* [41]
**Ca ^1^**	-	-	50
**K ^1^**	-	-	10
**Mg ^1^**	-	-	7.5
**P ^1^**	-	60	8.0
**Na ^1^**	-	8.0	6.8
**Fe ^1^**	-	-	0.5
**Zn ^1^**	-	-	0.5
**Mn ^1^**	-	-	2.0
**V ^2^**	-	-	5.0
**Cr ^2^**	-	-	500
**Co ^2^**	-	-	25
**Ni ^2^**	-	-	250
**Cu ^2^**	-	-	250
**As ^2^**	2.27	-	30
**Se ^2^**	-	-	3.0
**Mo ^2^**	-	-	100
**Ag ^3^**	-	-	100
**Cd ^2^**	2.27	-	10
**Hg ^2^**	0.11	-	0.2
**Pb ^2^**	11.36	-	10

*: Maximum tolerance levels of minerals in poultry.^1^ Values reported as g kg^−1^ DM. ^2^ Values reported as mg kg^−1^ DM. ^3^ Values reported as ng kg^−1^ DM.

**Table 6 animals-09-00189-t006:** Comparison of the rejection limits with obtained values for SAW 90 and 95 mix fed black soldier fly larvae.

	Estimated Value	Estimated Difference	*p*-value		Concentrations above Limits
			2-sided	1-sided (>0)	
**As**	2.71	0.000	0.001	<0.001	*
**Cd**	1.14	1.000	<0.001	1.000	
**Hg**	0.19	0.561	0.878	0.561	
**Pb**	0.33	1.000	<0.001	1.000	


Estimate refers to the estimated difference measured to the limit. Significant accumulation above the limits according to the one-sided test are denoted with *. No materials were found to be above the regulatory limits. Values reported as mg kg^−1^ DM.

**Table 7 animals-09-00189-t007:** Comparison of the maximum tolerance levels of metals with concentrations in SAW 90 and 95 mix fed larvae.

	Estimated Value	Estimated Difference	*p*-value		Concentrations above Regulatory Limits
			2-sided	1-sided (>0)	
**Ag** ^3^	0.01	−550.91	0.621	0.689	
**Ca** ^1^	75.36	74.81	<0.001	<0.001	*
**Co** ^2^	0.17	−550.75	0.621	0.689	
**Cr** ^2^	1.60	−549.31	0.622	0.689	
**Cu** ^2^	12.18	−538.74	0.629	0.685	
**Fe** ^1^	0.54	−0.01	0.993	0.504	
**K** ^1^	9.84	9.29	<0.001	<0.001	*
**Mg** ^1^	4.00	3.45	0.002	0.001	*
**Mn** ^1^	0.20	−0.35	0.755	0.623	
**Mo** ^2^	0.90	−550.02	0.622	0.689	
**Na** ^1^	2.03	1.48	0.186	0.093	
**Ni** ^2^	0.73	−550.19	0.622	0.689	
**P** ^1^	17.93	17.37	<0.001	<0.001	*
**Se** ^2^	1.12	−549.79	0.622	0.689	
**V** ^2^	0.85	−550.07	0.622	0.689	
**Zn** ^1^	0.26	−0.29	0.793	0.603	

Estimate refers to the estimated difference measured to the limit. Significant accumulation above the limits according to the one-sided test are denoted with *.^1^ Values reported as g kg^−1^ DM. ^2^ Values reported as mg kg^−1^ DM. ^3^ Values reported as ng kg^−1^ DM.

**Table 8 animals-09-00189-t008:** Bioaccumulation factor of different elements for black soldier fly larvae fed with SAW diets.

	SAW95	SAW90
Ag	0.3 ± 0.26	0.2 ± 0.16
As	0.9 ± 0.09	1.3 ± 0.08
Ca	1 ± 0.04	1.2 ± 0.31
Cd	2.5 ± 0.14	2.7 ±0.32
Co	0.4 ± 0.11	0.4 ± 0.09
Cr	0.4 ± 0.12	0.4 ± 0.13
Cu	0.7 ± 0.09	0.8 ± 0.1
Fe	0.8 ± 0.22	0.9 ± 0.14
Hg	1.6 ± 0.09	1.9 ± 0.17
K	9.1 ± 1.25	7.8 ± 2.16
Mg	1.1 ± 0.04	1.2 ± 0.27
Mn	1.3 ± 0.09	1.3 ± 0.1
Mo	0.6 ± 0.12	0.8 ± 0.12
Na	0.7 ± 0.09	0.7 ± 0.1
Ni	0.3 ± 0.12	0.4 ± 0.12
P	0.5 ± 0.08	0.6 ± 0.12
Pb	0.8 ± 0.12	0.8 ± 0.14
Se	0.5 ± 0.1	0.6 ± 0.1
V	0.3 ± 0.14	0.4 ± 0.13
Zn	0.5 ± 0.1	0.6 ± 0.1

SAW: Solid aquaculture waste. Values are mean ± standard deviation, calculated on dry matter basis. Values greater than 1 indicate accumulation.

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
