# Peer review of "Growth and Safety Assessment of Feed Streams for Black Soldier Fly Larvae: A Case Study with Aquaculture Sludge"

_animals, 2019, doi:10.3390/ani9040189_

Round 1

Reviewer 1 Report

Insects and products derived thereof are currently discussed as important food/feed sources. However, major safety concerns have been raised e.g. the accumulation of heavy metals. This manuscript reports the suitability of solid waste derived from aquaculture to breed black soldier fly larvae for subsequent use as food/feed. The main idea was to characterize the growth of the larvae together with the accumulation of unwanted metals according to EU-thresholds.

In principle this approach is valuable, but some general as well as specific weak points have to been dealt with. Such basic data may be suitable for a case study when further refined and more carefully presented.

More general criticisms: 

English should be improved by a native speaker.

The basic content of metals found in larvae of a control group fed conventional diets is missing.

Please use the term “crude protein” continuously when talking about N-containing substances detected by elemental analyser (therefore the protein content was only calculated, but not specifically measured using a distinct protein assay).

The obvious difference in “protein” concentrations between SAW90 and SAW95 is questionable and remains to be clarified (is there a loss of volatile N-substances?). 

Please explain why a 300°C treatment was introduced; due to EU-routine heat treatments up to 133°C are possible.

Each elements concentration should be given for the native SAW before heat treatments.

Specific points:

Figure 4: the classification of the x-axis remains doubtful because the experimental setup mentioned only 10 days; therefor the sentence in line 218+219 does not make sense. Furthermore “larval density” has not been defined and appears without unit; is “larval number” meant?

Line 75ff: sentence is misunderstanding because toxins may be reduced/eliminated during heat treatments.

Line 346ff: Such final sentences will not really help much in the sense of scientific added value.

Author Response

Dear Reviewer,

Thank you very much for your constructive remarks. We respond to your comments and the comments of the other reviewers in the attached PDF file. 

Thank you again,

The authors

Reviewer 2 Report

The results has to be improved and showed, not only by words, but also with table and graphs.

Author Response

(The authors gave the same response as above.)

Reviewer 3 Report

Manuscript Number: 464662 Animals

Dear Editor,

This paper of the title “Growth and safety assessment of feed streams for black soldier fly larvae: a case study with aquaculture sludge” reports on a study to investigate the growth performance and the accumulation of hazardous chemicals in BSFL fed with SAW based diets.

The manuscript is well written and the experiments appear to be well executed and the analysis is suitable. However, the topic is not very original.

I suggest the authors should implement several sections, adding information and rewrite the conclusions.

The main objection is that the authors performed all the experiments using two diets without a reference diet as control; they discussed their results only on the basis of data reported in others papers.

As for the BSFL growth performance, I suggest that more details on the fitness of the insects must be described. What is the mortality rate of the BSFL? Are they able to became pupae and thus adults? What is the fitness of these adults insects?

Minor comments:

- In the introduction section (line 70), the authors should refer to other recent papers about the microbiological safety of the diets for BSFL (e.g. Varotto Boccazzi et al, 2017 Plos One and Wynants et al 2018, Microb Ecol).

- M&M section: line 91, what is the time of the drying treatments? Please, add this information.

- Conclusion: lines 418-420: rewrite these sentences.

p { margin-bottom: 6.25px; line-height: 120%; }

Author Response

(The authors gave the same response as above.)

Round 2

Reviewer 1 Report

partly revised manuscript

Reviewer 3 Report

The reviewer appreciates the reply of the authors.